Vertical zonation of the Siberian Arctic benthos: bathymetric boundaries from coastal shoals to deep-sea Central Arctic

http://orcid.org/0000-0002-4457-8299 Vedenin Andrey 1 urasterias@gmail.com
http://orcid.org/0000-0002-9748-5168 Galkin Sergey 2
Mironov Alexander N. 2
http://orcid.org/0000-0001-7933-1651 Gebruk Andrey 2
1 Laboratory of Plankton Communities Structure and Dynamics, Shirshov Institute of Oceanology, Russian Academy of Sciences , Moscow, Moscow , Russia
2 Laboratory of Ocean Bottom Fauna, Shirshov Institute of Oceanology, Russian Academy of Sciences , Moscow, Moscow , Russia
Reimer James
Electronic publication date: 2021 Jun 29
Publication date: 2021
Volume: 9
Electronic Location ID: e11640
Received 2021 Mar 4; Accepted 2021 May 28
Copyright: © 2021 Vedenin et al.
Copyright year: 2021
Copyright holder: Vedenin et al.
License: This is an open access article distributed under the terms of the Creative Commons Attribution License, which permits unrestricted use, distribution, reproduction and adaptation in any medium and for any purpose provided that it is properly attributed. For attribution, the original author(s), title, publication source (PeerJ) and either DOI or URL of the article must be cited.
License URL: https://creativecommons.org/licenses/by/4.0/

Keywords: Biogeography, Arctic benthos, Vertical zonation, Siberian seas

Funding: Russian Foundation for Basic Research (RFBR) 18-05-60228 This work was supported by the the Russian Foundation for Basic Research (RFBR) project (No. 18-05-60228). The funders had no role in study design, data collection and analysis, decision to publish, or preparation of the manuscript.

==============================
The bathymetric distribution of species of Annelida, Crustacea and Echinodermata from the region including the Kara, Laptev and East Siberian seas and the adjacent region of the deep-sea Central Arctic was analysed. We focused on vertical species ranges revealing zones of crowding of upper and lower species range limits. Using published data and in part the material obtained during the expeditions of the P.P. Shirshov Institute of Oceanology, we evaluated species vertical distribution from 0 m to the maximum depth of the Central Arctic (~4,400 m). The entire depth range was divided into smaller intervals; number of upper and lower limits of species depth ranges was counted and plotted to visualize the range limits crowding. Several zones of crowding of vertical species range limits were found for all analysed macrotaxa. The most significant zones occurred at depths of 450–800 m and 1,800–2,000 m. The first depth zone corresponds to the boundary between the sublittoral and bathyal faunas. The last one marks the boundary between the bathyal and abyssal faunas. Depths of these boundaries differ from those reported from other Ocean regions; possible explanations of these differences are discussed.

Introduction

Vertical zonation of marine fauna has been the subject of repeated interest over many years. In published schemes of vertical zonation many authors recognised the major boundary at ~200 m, the border between the sublittoral and the bathyal zoned (Belyaev et al., 1959; Vinogradova, 1962; Carney, Haedrich & Rowe, 1983; Mironov, 1986; Howell, Billett & Tyler, 2002; Thistle, 2003; Carney, 2005; Jamieson, 2015). Watling et al. (2013) re-examined the biogeographic classification of depth ranges in the deep sea and suggested that this boundary lies deeper, at 300 m. The second major vertical division is the border between the bathyal and abyssal. In the scheme of the vertical zonation suggested by Belyaev et al. (1959), the bathyal is defined as a depth zone between 200 m and 3,000 m, with a zone between 2,500 and 3,500 m considered as the transition between the bathyal and abyssal zones. The boundary at ~3,000 m was demonstrated in a number of publications (Vinogradova, 1962; Krayushkina, 2000; Howell, Billett & Tyler, 2002; Gebruk, Budaeva & King, 2010), however some authors recognised the border between the bathyal and abyssal closer to 2,000 m (Gage & Tyler, 1992; Thistle, 2003) or 4,000 m (Thurman, 1985). In classification of Watling et al. (2013), the “upper bathyal” at 300–800 m and the “lower bathyal” at 800–3,500 m were reported as separate zones.

Schemes of vertical zonation with universal boundaries at the same depth throughout the Ocean indicate only “average” depths of boundaries (Hedgpeth, 1957; Belyaev et al., 1959; Vinogradova, 1962; Gage & Tyler, 1992; Thistle, 2003; Watling et al., 2013). In reality, depths of vertical zones significantly vary from one region to another. Thus, boundaries at different depths depending on a region were shown for a number of macrotaxa based on the same method. Among the examples are the asellote isopods (Menzies, George & Rowe, 1973) and the echinoids (Mironov, 1986), though these studies were performed outside the Arctic. The Arctic Ocean markedly differs from other areas in the vertical biodiversity trends. Outside the Arctic, a parabolic trend of benthic diversity is observed with peak values at ~2,000–3,000 m (Rex & Etter, 2010). In the Kara, Laptev and East-Siberian Seas the peak of species richness occurs at approximately 100 m, in the Barents Sea—slightly deeper 100 m (Anisimova, 1989; Stepanjants, 1989; Vassilenko, 1989; Vedenin et al., 2018), whereas in the Chukchi Sea—shallower 100 m (Golikov, 1989; Stepanjants, 1989; Vassilenko, 1989).

Schemes of vertical zonation are not always comparable because methods and approaches used by different authors vary significantly and sometimes fundamentally. Moreover, approaches to zonation are not always clearly explained. The variety of approaches to define vertical zones in the Ocean was examined and structured by Golikov et al. (1990) and Mironov (2013), who distinguished three main approaches to marine biogeography differing in the subject of study and methods. The first approach deals with species range limits, named “biotic” (or “faunistic”) approach, according to classification of Mironov (2013). The second one is the “biocenotic” approach, which deals with communities (or biocenoces). The third approach is the “landscape”, which considers environmental parameters.

According to Mironov (2013), in the biotic approach the main criterion of a biogeographic boundary (=biotic boundary) is a zone of crowding of species range limits. In case of vertical zonation, depth ranges of crowding of upper or lower limits of species vertical ranges are considered. This approach was used in different Ocean regions on the example of various invertebrate macrotaxa and fish (Backus et al., 1965; Carney, Haedrich & Rowe, 1983; Gage et al., 1984; Mironov, 1986; Gage & Tyler, 1992). In the biocenotic approach the criterion of a boundary is a difference between local biotas commonly revealed using cluster analysis (based on such parameters as species diversity, abundance and biomass) (Longhurst, 1985; Grassle & Morse-Porteous, 1987; Rex & Etter, 2010). In the landscape approach boundaries usually correspond to sharp gradients of physical-chemical parameters (Milkov, 1970; Deacon, 1982; Gukov, 1999; Watling et al., 2013).

In the Siberian Arctic, most of previous studies of vertical zonation of benthic fauna were based on the biocenotic (community-based) or mixed (based on methods with certain elements of biotic, biocenotic and landscape) approaches. Published data mainly relate to the Kara Sea (Filatova & Zenkevich, 1957; Antipova & Semenov, 1989; Jørgensen et al., 1999; Anisimova et al., 2003; Deubel et al., 2003; Galkin & Vedenin, 2015; Vedenin, Galkin & Kozlovskiy, 2015), in part to the Laptev Sea (Fütterer, 1994; Sirenko et al., 2004), the East Siberian Sea (Gukov et al., 2005; Sirenko & Denisenko, 2010) with a few studies dealing with the deep-sea Central Arctic (Deubel, 2000; Vedenin et al., 2018; Rybakova et al., 2019).

The Norwegian and Greenland seas are the closest to the Siberian Arctic where the biotic (species-based) approach was applied (Zhirkov & Mironov, 1985; Svavarsson, Brattegard & Strömberg, 1990; Krayushkina, 2000; Oug et al., 2017). Several zones of crowding of bathymetric limits were shown: at depths 450–700 m, 900–1,000 m and ~2,000 m for Annelida (Zhirkov & Mironov, 1985; Oug et al., 2017), 150–200 m and 1,100–1,200 m for Asteroidea and 600–800 m for Holothuroidea (Krayushkina, 2000). These depth ranges (except 150–200 m fro Asteroidea) differ from those shown for other parts of the Ocean. However, even in the Norwegian Sea (one of the best studied regions of the Arctic Ocean) the zones of crowding matched depth ranges with the highest number of samples (Zhirkov & Mironov, 1985) suggesting a bias related to the sampling effort.

In the present study we tried to collect all available data for the Siberian Arctic on the bathymetric distribution of three macrotaxa: Annelida, Crustacea and Echinodermata. We aimed at examining the upper and lower vertical limits of species ranges (in these macrotaxa) to reveal the depth of limits crowding marking potential biogeographic boundaries. We hypothesized that the boundaries between the sublittoral and bathyal, and between the bathyal and abyssal faunas exist in the Arctic, but are at located at depths different from those in other ocean areas.

Materials & methods

The study area included the Kara Sea, Laptev Sea, East Siberian Sea and the adjacent sector of the Central Arctic (Fig. 1). The area marked as “the Central Arctic” in Fig. 1 is bordered by dashed lines since we also considered species recorded from adjacent areas owing to supposed uniformity of fauna of central Arctic basins ((Bluhm et al., 2010).

Figure 1 Study area with formal borders of the Kara, Laptev and East Siberian seas.

The adjacent sector of the Central Arctic Basin is outlined with dashed lines—the uniformity of fauna of the entire deep-sea Central Arctic is suggested.

Data on the bathymetric ranges of Annelida, Crustacea and Echinodermata were previously published in Data in Brief (Vedenin, Galkin & Gebruk, 2021). From all listed species in these three macrotaxa, we selected about two thirds “most reliable”: 166 from 253 species of Annelida; 372 from 464 species of Crustacea and 51 from 63 species of Echinodermata. The selection was based on the following criteria:Species distribution. Species known only from borders of basins (e.g. from the western-most Kara Sea or the eastern-most East Siberian Sea) were excluded from the analysis.

Number of findings. At least two known depth records from the study area were required.

Taxonomy problems. Taxa with questionable taxonomical status (e.g. taxon inquirendum or unresolved species complexes) were excluded from the analysis.

Overall reliability of published information. We used mostly publications focused on the taxonomy, since there is a higher likelihood of identification mistakes in ecological publications.

The complete species list used for the analysis is presented in the Supplemental 1. The procedure we used after compiling the list of species records was based on methods described by Vinogradova (1962) and Mironov (1986, 2013). Important characteristics of the biotic approach are the total exclusion of environmental data from primary analysis and the assumption of the continuity of a species range (including the bathymetric range). Disjunctions in the distribution of species are disregarded (Mironov, 2013).

The entire depth range of sampling (from 0 to 4,400 m) was divided into 200-m intervals. In addition, the upper 800 m horizon was divided into 50-m intervals. The number of upper and lower limits of species depth ranges within each 200 m- or 50 m-interval was counted and plotted as a linear graph.

The observed number of vertical limits was compared to the expected one, calculated using the equation suggested by Backus et al. (1965). In the original paper, the vertical zonation of mesopelagic fishes was studied based on the Isaacs-Kidd trawl samples at subsequent depth ranges. The Backus model proposes that the expected number of species in each of the subsequent depth intervals will remain constant if no biogeographic boundaries are present (Backus et al., 1965; Gage, 1986). In the upper-most depth range (the first haul) all species will be met the first time (the number of species sampled = the number of upper limits within the upper depth range). The likelihood of meeting for the last time each sampled species will be increasing respectively in each of the next, deeper depth ranges. The Backus regularity can be expressed as the following equation: as(x)=k−kN∑n=0x−1⁡as(n), where as(x)—number of upper species limits within the x depth interval; k—number of species occurring in each x depth interval presumed in the model as constant; N—total number of species, occurring in the entire depth range; ∑n=0x−1⁡as(n)—sum of the upper limits, calculated in previous depth intervals. The lower limits of species distribution can be calculated similarly, though their number is conversely increasing with depth, and all sampled species within the last (the deepest) depth range will represent the lower limits. As an example, imagine if it was a total of 1,000 species within the 3,000 m depth range divided into 200-m intervals with ~200 species occurring in each of the intervals. The examples of expected upper and lower limits of distribution calculated using these values are shown in Fig. 2. Direct calculations behind the examples are available in Supplemental 2.

Figure 2 Example of the plotted expected number of upper and lower vertical limits of species distribution based on Backus equation.

a(x) upper—expected number of the upper vertical limits; a (x) lower—expected number of the lower vertical limits; a(x) upper + a (x) lower—summarized values of upper and lower vertical limits.

The expected number of upper and lower limits of species distribution was then summarized (Fig. 2). The difference between the observed and expected number of limits was assessed using the Chi-square test. Differences were considered reliable if the Chi-square results were >3.84 (degree of freedom = 1; p-value = 0.05) where the observed number of limits overreached the expected number of limits (Franke, Ho & Christie, 2012).

For the details of the described method see Backus et al. (1965), Gage et al. (1984), Gage (1986) and Bamber & Thurston (1995).

The vertical distribution of benthic stations taken between the years 1881 and 2015 was analysed to avoid the bias related to uneven distribution of stations (and samples) within the study area. A set of 1,046 stations was tested; the complete list with expedition data and coordinates is shown in Supplemental 3. Spearman ranked correlations were calculated between the stations and species distribution along different depth ranges.

Results were plotted as simple graphs, using the Microsoft Office software. Maps were built using the Ocean Data View 5.3.0 software (Schlitzer, 2020).

Results

Annelida distribution

Most of the Annelida species occurred within the upper hundreds of meters. Reliable concentrations of the upper and lower range limits were found within the depth ranges of 450–600 m (mean Chi sq. value—38.7) and 700–750 m (mean Chi sq. value—31.5). Those zones of crowding were revealed applying 50 m increments in the upper 800 m (Fig. 3A).

Figure 3 Distribution of upper and lower vertical limits in species of Annelida.

Observed limits are coloured in blue; expected—in red; grey bars indicate Chi sq. values. (A) Upper 800 m divided into 50-m intervals; (B) Entire depth range divided into 200-m intervals.

Along the entire depth range, zones of crowding of vertical limits occurred at 1,000–1,200 m (16 limits, Chi sq. value—9.9), 1800-2000 m (5 limits, Chi sq. value—5.0) and 2,200–2,400 m (5 limits, Chi sq. value—6.9) (Fig. 3B).

Crustacea distribution

Different taxa of Crustacea are characterized by different migration and evolutionary history and show unequal diversity along the depth gradient in the Arctic Ocean (Gurjanova, 1951; Brandt et al., 1996; Brandt, 1997). Therefore, the following Crustacea taxa were analysed separately: Cirripedia, Decapoda, Cumacea, Mysida, Tanaidacea, Isopoda and Amphipoda.

In the first five taxa no significant zones of crowding of the upper and lower vertical limits were found. The species number within each taxon was too small to build a reliable plot (Table 1).

Table 1 Summarized upper and lower species vertical limits in different taxa of Crustacea per 200-m depth increments.

Crustacea taxon	Number of upper and lower vertical limits within 200-m depth intervals	
	0–199	200–399	400–599	600–799	800–999	1000–1199	1200–1399	1400–1599	1600–1799	1800–1999	2000–2199	2200–2399	2400–2599	2600–2799	2800–2999	3000–3199	3200–3399	3400–3599	3600–3799	3800–3999	4000–4199	4200–4399	
Cirripedia	6	1	1	2		2			1			1											
Decapoda	13	4	6	2	4			2				2			1			1				1	
Mysidacea	14	3	2	2	1	2		4		1		2	1			2		1				1	
Cumacea	40	9	7	3	2	2	1		1		1	3	1		1			1					
Tanaidacea	6		1				1			2	2					1	3						
																							
	Number of upper and lower vertical limits within 50-m depth intervals (the upper 800 m)	
	0–49	50–99	100–149	150–199	200–249	250–299	300–349	350–399	400–449	450–499	500–549	550–599	600–649	650–699	700–749	750–799	
Cirripedia	6							1			1				2		
Decapoda	11	1			1	4				3	3			1	1		
Mysidacea	10	2		2	2	1			1	1				1		1	
Cumacea	30	7	1	1	4	3		2	1	4	1	1		2	1		
Tanaidacea	4	1	1									1					

In Isopoda significant concentrations of the upper and lower vertical limits were revealed at depth layers 500–700 m and 1,200–1,400 m (Chi sq. values 18.2 and 16.8, respectively) (Fig. 4). Peaks at 500–700 m were mostly related to the upper vertical limits (11 out of 17 in total), whereas peaks at 1,200–1,400 m corresponded to the lower vertical limits (10 out of 11 in total).

Figure 4 Distribution of upper and lower vertical limits in species of Isopoda.

Observed limits are coloured in blue; expected—in red; grey bars indicate Chi sq. values. (A) Upper 800 m divided into 50-m intervals; (B) Entire depth range, divided into 200-m intervals.

In Amphipoda concentrations of vertical limits appeared at 450–550 m (mean Chi sq. value—4.4), 650–700 m (Chi sq. value—4.1) and 1,800–2,000 m (Chi sq. value—6.2). Statistically unreliable small peak was observed at 1,000–1,200 m depth (Chi sq. value < 3.8) (Fig. 5).

Figure 5 Distribution of upper and lower vertical limits in species of Amphipoda.

Observed limits are coloured in blue; expected—in red; grey bars indicate Chi sq. values. (A) Upper 800 m divided into 50-m intervals; (B) Entire depth range, divided into 200-m intervals.

Unlike in Isopoda, most of these zones of crowding corresponded to the lower limits of species vertical ranges.

Echinodermata distribution

In Echinodermata, similar to Annelida and Crustacea, statistically reliable peaks of of vertical limits concentration were revealed at 450–600 m depth (Chi sq. value—4.3). Additional zone of crowding occurred at 2,200–2,400 m (Chi sq. value—6.5) and two unreliable concentrations were observed at 1,000–1,200 m and 1,800–2,000 m (Chi sq. values < 3.8) (Fig. 6).

Figure 6 Distribution of upper and lower vertical limits in species of Echinodermata.

Observed limits are coloured in blue; expected—in red; grey bars indicate Chi sq. values. (A) Upper 800 m divided into 50-m intervals; (B) Entire depth range, divided into 200-m intervals.

These peaks, like in Amphipoda and Annelida, mainly corresponded to the lower vertical limits—10 out of 11 in total at 450–600 m; 6 out of 6 at 1,000–1,200 m; 3 out of 3 at 1,800–2,000 and 4 out of 4 at 2,200–2,400 m.

Station distribution

Analysed stations were distributed unevenly along the depth. Most of the stations were taken on the shelf within the upper 100 m (389 out of 1,046 stations). Deeper 200 m, the number of stations per each 200-m interval was relatively small (Fig. 7).

Figure 7 Distribution of stations by depth.

(A) Upper 800 m divided into 50-m intervals; (B) Entire depth range, divided into 200-m intervals.

The distribution of stations along the depth in general correlated with most of observed patterns of taxa vertical distribution. One exception was the vertical distribution of Amphipoda and Echinodermata calculated per 200-m depth intervals (Table 2). However, despite the overall correlation, certain peaks of vertical limits in Annelida, Crustacea and Echinodermata did not correspond to the peaks in the distribution of stations, for example at 600–750 m in Isopoda and Amphipoda, at 1,200–1,400 m in Isopoda and at 1,800–2,000 m in Annelida, Amphipoda and Echinodermata.

Table 2 Spearman ranked correlation values between the number of stations taken at different depths and the number of upper and lower vertical limits in species of Annelida, Crustacea and Echinodermata.

Taxon, correlated with number of stations	50-m intervals for upper 800 m	200-m intervals for entire depth range	
R	p	R	p	
Annelida	0.60	0.0137	0.43	0.0449	
Isopoda	0.52	0.0380	0.48	0.0222	
Amphipoda	0.74	0.0010	0.41	0.0577	
Echinodermata	0.58	0.0197	0.27	0.2273	
SUM	0.69	0.0029	0.52	0.0131	

Revealing the bathymetric boundaries

Combined data for Annelida, Crustacea and Echinodermata demonstrated several distinct zones of crowding of the upper and lower species vertical limits (Fig. 8). Peaks occurred in all taxa at the following depths: 450–600 m, 650–700 m (Fig. 8A), 1,000–1,200 m, 1,800–2,000 m and 2,200–2,400 m (Fig. 8B). Annelida and Amphipoda, as more species rich, contributed more significantly to this pattern.

Figure 8 Combined distribution of the upper and lower species vertical limits in Annelida, Crustacea and Echinodermata.

Observed limits are coloured in blue; expected—in red; grey bars indicate Chi sq. values. (A) Upper 800 m divided into 50-m intervals; (B) Entire depth range, divided into 200-m intervals.

The boundary at 1,000–1,400 m was clearly pronounced only in annelids and isopods, however, it was also visible in amphipods and echinoderms. At greater depths, the clearest concentrations of vertical limits occurred at 1,800–2,000 m in annelids and amphipods and less pronounced at 2,200–2,400 m. At this depth the lower vertical limits of several bathyal species occurres, such as the ophiuroid Ophiopleura borealis and the asteroids Bathybiaster vexillifer and Pontaster tenuispinus (Supplemental 1).

Discussion

A number of schemes of vertical zonation based on the distribution of benthic fauna in various Arctic regions was suggested earlier (Mironov, 2013). These schemes are barely comparable owing to the difference in methods and approaches used. The number of biogeographic regions or vertical zones increases from schemes based purely on the biotic approach (species—based) to those based on mixed elements of the biotic and biocenotic (communities—based) approaches, and increases further in schemes based on the mixed biocenotic-landscape approaches (Mironov, 2013). Below we consider only the schemes of vertical zonation based on the biotic approach (Zhirkov & Mironov, 1985; Mironov, 1986; Gage, 1986; Svavarsson, Brattegard & Strömberg, 1990; Howell, Billett & Tyler, 2002; Oug et al., 2017) (Fig. 9).

Figure 9 Comparison of vertical boundaries in Annelida, Crustacea and Echinodermata revealed in our study with published data on other regions.

Black colour indicates significant concentrations of the upper and lower vertical limits of species ranges. Grey colour indicates statistically insignificant concentrations.

The true biotic approach to reveal vertical zones in the Arctic Ocean basin has been applied only in the Norwegian Sea (Zhirkov & Mironov, 1985; Krayushkina, 2000). Additionally the distribution and diversity patterns of the asellote isopods in the deep Norwegian and Greenland Seas were studied by Svavarsson, Brattegard & Strömberg (1990). The authors used the cumulative number of first occurrences of species, the approach similar to the biotic one. Oug et al. (2017) examined changes with depth of the species richness of annelids in the deep Nordic Seas at the alpha (sample species richness), beta (turnover) and gamma (large area species richness) scales. Menzies, George & Rowe (1973) studied the vertical distribution of assellote isopods in the high Arctic. The authors suggested a simple method of determining the faunal homogeneity or distinctiveness between depth intervals: the total taxa in common (Tc) between adjacent depth intervals was subtracted from the total taxa (T) at the two depth intervals, divided by the total (T) and multiplied by 100 to gain the percentage of distinctiveness (D): T – Tc / T × 100 = distinctiveness D.

The boundary at 450–800 m revealed in our study for the first time roughly corresponds to boundaries shown in Krayushkina (2000) and Oug et al. (2017) (Fig. 8) and is close to the boundary at 425 m in Menzies, George & Rowe (1973). The boundary at 1,200–1,400 m (for Isopoda in our study) also can be found in Krayushkina (2000), whereas the boundary at 1,800–2,000 m is close to the boundary in the scheme of Oug et al. (2017). Our results do not correlate with the schemes of Zhirkov & Mironov (1985) and Svavarsson, Brattegard & Strömberg (1990) (Fig. 9), however in the later scheme there is a boundary at 2,000 m revealed using different methods (cluster analysis—the biocenotic approach).

We suggest that the depth interval 450–800 m in our study represents the boundary between the sublittoral and bathyal faunas since it was the shallowest reliable boundary we found. Such biogeographic boundaries are associated with sharp breaks in gradients of environmental parameters (Rapoport, 1982; Mironov, 2004; Mironov, Dilman & Krylova, 2013; Watling et al., 2013). In case of the sublittoral zone, its lower boundary is usually associated with pronounced changes in the deposition and burial of the organic matter at the seafloor. The lower boundary of the faunistic sublittoral zone roughly corresponds to the lower boundary of the photic zone and the depth of the continental shelf break in the Ocean at ~200 m (Ekman, 1953; Carney, Haedrich & Rowe, 1983; Mironov, 1986; Carney, 2005; Wei et al., 2010). In particular, the exact depth of this boundary was reported at 200–300 m off the North-East coast of the USA (Haedrich, Rowe & Polloni, 1980; Rowe, Polloni & Haedrich, 1982); at 150–200 m in the North-East Atlantic (Mironov, 1986). Overall, the lower sublittoral boundary is usually drawn at depths shallower 400 m (Hedgpeth, 1957; Belyaev et al., 1959; Vinogradova, 1962; Haedrich, Rowe & Polloni, 1980; Gage & Tyler, 1992; Thistle, 2003; Watling et al., 2013).

However, in the Arctic seas conditions for photosynthesis are different from the lower latitudes since they are constrained by the ice cover and the lighting regime (Boetius et al., 2013; Flint et al., 2019). By generalised characteristics of photosynthesis, the shelf of the Siberian seas resembles the so called «mesophotic ecosystems». Such ecosystems are characterized by low-light conditions (Hinderstein et al., 2010; Easton et al., 2019; Pyle & Copus, 2019; Mecho et al., 2021). Apparently the break in the photosyntetic gradient in the Siberian seas at depths about 200 m is weakly pronounced. If so, it can be suggested that the biotic boundary has not developed there owing to the lack of sharp changes at this depth in the deposition of organic matter of photosynthetic origin to the seafloor in the Siberian seas.

The revealed in this study deeper sublittoral boundary at 450–800 m apparently indicates significant changes in trophic conditions at this depth. However, specific environmental factors controlling the inflow of organic matter at this depth are not obvious, further studies are required. The “deep” lower sublittoral boundary is also a feature of the Antarctic (Fig. 9, Mironov, 1986). It was suggested that this effect in the Antarctic is related to the peculiar geomorphology with the continental shelf sunken as a result of the glaciers pressure (Brey et al., 1996; Thatje, Hillenbrand & Larter, 2005; Martín-Ledo & López-González, 2014). During the series of the Pleistocene glaciations, the ice sheet was bending the Antarctic crust forcing many shallow-water species to migrate deeper and adapt to eurybathy (Brey et al., 1996; Thatje, Hillenbrand & Larter, 2005). However, this is not the case for the Arctic Ocean, since the history of the Arctic fauna is completely different. Within the study area, possible reasons for the described boundary can be associated with peculiar geomorphology of the shelf-slope area and the hydrological regime. The Siberian shelf is one of the widest in the world, up to 800 km (Weber, 1989). In addition, it is characterized by a strong fresh water outflow (Flint et al., 2019). As an example, the annual freshwater discharge from all major Arctic rivers is approximately 3,300 km3, equivalent to 10% of the global river-off. Just two rivers, the Ob and Yenisei, discharge about 30% of the total annual river run-off into the Arctic Ocean through the Kara Sea (Fütterer & Galimov, 2003; Stein, 2000). From the ocean side, the upper slope of the Siberian seas is bathed by the warm Atlantic waters (Aagaard, 1989; Rudels et al., 1994; Bluhm et al., 2020), the factor that potentially can influence trophic conditions by enriching the upper slope with organic carbon (Dunton, 1992; Wassmann, Slagstad & Ellingsen, 2019; Bluhm et al., 2020). The lower boundary of the Atlantic layer in the Siberian Arctic represents a rather sharp gradient from 2–3 °C and ~35 psu to ~−1 °C and ~32–33 psu at depths of 500–700 m (Bluhm et al., 2020). Faunal changes coinciding with the inflow of Atlantic waters were reported for annelids, isopods, gastropods and fishes in the Norwegian Sea, where the gradient is particularly high (Svavarsson, Brattegard & Strömberg, 1990; Bergstad, Bjelland & Gordon, 1999; Høisæter, 2010; Oug et al., 2017).

The boundary between the bathyal and abyssal faunas revealed in our study for the first time at the depth of 1,800–2,000 m lies, on the contrary, shallower than in many other ocean regions (~3,000 m) (Gebruk, Budaeva & King, 2010). The bathyal-abyssal fauna boundary is considered by some authors as related to the decrease in food availability with depth, certain changes in temperature regime (such as the 4 °C isotherm) and the near-bottom currents associated with changes in the bottom topography at the transition from the continental slope to the continental rise and abyssal plain at 2,400–3,600 m (Menzies, George & Rowe, 1973; Gage, 1986; Gage & Tyler, 1992; Thistle, 2003; Wei et al., 2010; Watling et al., 2013). The decrease in the food availability with depth demonstrated rapid drops in benthic production approximately within the continental rise area in the Central Arctic (Degen et al., 2015). Important feature of the continental rise is that it represents the site of accumulation of most of the detrital sediment eroded off the continents (Hay, 2016). Sokolova (2000) presented evidence that at the transition from the continental rise to the continental slope in the Ocean (at ~3,000 m depth) the organic carbon content in the sediment is higher. This is a result of accumulation of organic matter buried in the sediment sliding down the slope (Hay, 2016). This zone contours all continents and it was marked by Sokolova (2000) as the “Circumcontinental eutrophic zone”. In the Arctic Ocean the decrease of the slope inclination at the transition from the continental shelf to the continental rise was shown for some areas north of the Kara, Laptev and East-Siberian seas at ~2,000 m (Jakobsson et al., 2020; Bluhm et al., 2020). This is shallower than in the scheme of Sokolova (2000) and this is apparently a regional peculiarity of the basin. Certain changes in benthic communities were revealed using the biocenotic approach in some areas of the Laptev and East-Siberian seas at ~1,900–2,000 m (Sirenko et al., 2004) and 1,740–2,100 m (Deubel, 2000). There are reasons to believe that these changes and the boundary identified in the present study at 1,800–2,000 m are related to the bottom topography and first of all to the transition from the continental rise to the continental slope at this depth.

Whatever the case, the bathymetric distribution of benthic fauna in the Arctic Ocean has unique features not found in other Ocean regions. The generalized scheme of vertical zonation of the Arctic benthic fauna based on the analysis of species range limits is shown in Fig. 10.

Figure 10 Scheme of vertical zonation of benthic fauna distribution in the Siberian Arctic (based on species vertical ranges).

Examples of characteristic species: 1—Branchiomma arctica; 2—Leptasterias groenlandica; 3—Ampelisca macrocephala; 4—Neohela monstrosa; 5—Melinnopsis arctica; 6—Bathybiaster vexillifer; 7—Elpidia heckeri; 8—Ymerana pteropoda; 9—Liljeborgia polosi. (Photographs of A. Vedenin). Dashed lines indicate approximate depths of corresponding boundaries outside the Arctic Ocean: upper line (green)—the boundary at ~200 m (sublittoral/bathyal), bottom line (yellow)—the boundary at ~3,000 m (bathyal/abyssal).

Conclusions

The boundaries between the sublittoral, bathyal and abyssal faunas exist in the Arctic, but they lie at depths uncommon for other Ocean areas. Reported in the present study the zone of crowding (concentration) of the upper and lower species vertical limits at 450–800 m likely marks the boundary between the sublittoral and bathyal faunas. Commonly in the Ocean this boundary occurs at 150–250 m depth (except for the Antarctic). It remains unclear why in the Siberian seas the lower boundary of the sublittoral zone lies deeper than elsewhere in the Ocean. The boundary between the bathyal and abyssal faunas was observed at 1,800–2,000 m, which is, on the contrary, shallower, than in other Ocean regions (for example 2,400–3,600 m in the North Atlantic and North Pacific). It can be suggested that this boundary at least in part is a result of the transition from the continental slope to the continental rise at shallower depths (~2,000 m) in the Siberian Arctic.

Supplemental Information

Supplemental Information 1 List of taxa used in this study.

List of species of Annelida, Crustacea and Echinodermata with the shallowest and deepest known findings within the study area and corresponding references where the depths were taken.

Click here for additional data file.

Supplemental Information 2 Calculations of the expected number of vertical limits using Backus equation.

The example of the Backus model with the expected number of upper, lower and summarized upper and lower vertical limits; calculation of the expected number of vertical limits for Polychaeta; calculation of the expected number of vertical limits for Isopoda; calculation of the expected number of vertical limits for Amphipoda; calculation of the expected number of vertical limits for Echinodermata.

Click here for additional data file.

Supplemental Information 3 List of benthic stations with expedition details, coordinates and depth used for this study.

Click here for additional data file.

The authors are thankful to Dr. Natalya Budaeva, Dr. Kirill Minin, Dr. Alexey Udalov, Dr. Vassily Spiridonov and Dr. Victor Petryashev for their help with literature search. Many thanks are due to Dr. Antje Boetius and Dr. Ingrid Krönke for providing original macrobenthic samples from earlier expeditions.

Additional Information and Declarations

Competing Interests

Author Contributions

Data Availability

The authors declare that they have no competing interests.

Andrey Vedenin conceived and designed the experiments, performed the experiments, analyzed the data, prepared figures and/or tables, authored or reviewed drafts of the paper, and approved the final draft.

Sergey Galkin performed the experiments, authored or reviewed drafts of the paper, and approved the final draft.

Alexander N Mironov conceived and designed the experiments, performed the experiments, authored or reviewed drafts of the paper, and approved the final draft.

Andrey Gebruk analyzed the data, authored or reviewed drafts of the paper, supervising the work, and approved the final draft.

The following information was supplied regarding data availability:

The list of species, details of statistical analysis and the set of benthic stations are available in the Supplemental Files.

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
