# Peer review of "Vertical zonation of the Siberian Arctic benthos: bathymetric boundaries from coastal shoals to deep-sea Central Arctic"

_PeerJ, doi:10.7717/peerj.11640_

## Round 0.1 · original submission · Major Revisions

I have heard back from two reviewers, both of whom offer very constructive comments. Reviewer 1 notes that your discussion needs more work, as you do not highlight well what is new about your work. Reviewer 2 has many questions about your statistical analyses. Taken together, answering these and the other comments will require a major revision.

Reviewer 1 ·

Basic reporting

This study reviews the vertical distribution of organisms in the Arctic Ocean based on several previous studies and identifies the biota's boundaries. The methodology itself appears to be sound. The information contained in the paper is valuable and deserves publication.
However, some improvements could be made to the study.

1. Discussion
Most of the discussion merely traces previous research, making it difficult to understand what this study has revealed.
Since you have reviewed the biota based on many previous studies, it would be good to discuss each bathymetric zone's characteristics in more detail based on its constituent species.

Since you have reviewed the biota based on many previous studies, it would be good to discuss each bathymetric zone's characteristics in more detail based on its constituent species.
There is still a lot to discuss, such as why the species are present in that depth zone and whether changes in species composition are common among the three phyla.

You mentioned the Antarctic Ocean, but little about the rest of the world. I think that you should clarify the characteristics of the study area compared to previous studies on the vertical distribution in other parts of the world.

In fact, it would have been better to include some basic information, such as water temperature, which is easily available, to add more depth to the discussion. Still, I leave that to the author since the author declares that he will not do so at the beginning of the discussion.

2. List of species
This is inevitable, given the large number of previous studies reviewed and the wide range of taxa covered, but some errors are apparent.
e.g.
Polychaeta should not be used. Annelida is better.
Travisia is not Opheliidae (now Travisiidae).
We recommend that you check again as there are likely to be many other errors.


Thank you for inviting me to review,
Yours sincerely.

Experimental design

no comment

Validity of the findings

no comment

Additional comments

no comment

Reviewer 2 ·

Basic reporting

The paper is moderately well written. There are typos and missing words throughout, some of which I expect are due to the translational thinking from Russian to English.

Literature is sufficient, but some additional statistical references might be added re ChiSquare

Experimental design

The major problem I have with this paper is the explanation of the statistical analyses.

I have been puzzling over the use of Chi-square as the test distribution and the parabolic shape of the expected values in Figures 2-7. If one looks only at the values for upper and lower limits within a depth range (the blue line) one can see that the values don’t change much, e.g., in Fig. 7, below a depth of 1400 m. Yet, the expected values soon bottom out and then start to go way up, as do the Chi-square values.
l. 145-148 merely states that the expected number of upper and lower limits is summarized. There is no statement in the previous paragraphs about how the expected number of species for each depth interval is derived. One can only guess.
Why do the expected values rise so rapidly with depth? If the class interval expected values are not all equal, then some underlying distribution must have been applied. One thing we know for Chi-square, is that it is notoriously unreliable when values are small, i.e., below 10, and in the case where some cell values are very high and others very low, the cells with low numbers are often combined. In this case it seems that the number of upper and lower limits are routinely pretty low.

Validity of the findings

A few additional comments:

l. 160-165 and elsewhere for the other taxonomic groups…. Is it really a zone of crowding when you have only a small number of limits, e.g., 5 to 10, when the overall polychaete species richness must be quite a bit larger. In other words, if a good survey of the benthos at those depths would produce 50-100 species of polychaetes, then having 5-10 species define a zone of crowding doesn’t make sense since those are only 10% or so of the total.
l. 190-192, as you note, those peaks apply mainly to lower limits. That is not a zone of crowding, but rather a slow depletion of species who have met conditions in which they can no longer exist. One would expect that sort of winnowing with depth as food is steadily being depleted.

Additional comments

In sum, I think the methodological aspects of the calculations need more thorough explaining. I tried finding the Backus paper on the internet, both through Harvard, who said go to Biodiversity Heritage Library, a fabulous site, that alas did not have this volume of the Bull MCZ. So I cannot figure out how the calculations are made, as noted previously.
Because of the low number of records with depth, I am not so sure that Chi-square is the statistic to use. There are some very good books on dealing with categorical data, but unfortunately I am not near my office or I would make some recommendations.
Lastly, I wonder if taking the biocoenosis approach would also be useful as a comparison to the approach used here. I suspect in many ways the limited data might also be a problem for the biocoenotic approach as well.

---

## Round 0.2 · Minor Revisions

Thank you very much for your revision; there are just two small issues to attend to, and I imagine you can complete this revision quickly.

Reviewer 1 ·

Basic reporting

The manuscript has been revised on the basis of the reviewers' comments and no further comments have been made on my part.
Minor corrections
1. Author should not use "Polychaeta" in the manuscript not only in the excel file.
2. Vedinin et al. (2021) is already open in the journal site as a pre-proof. The author should add it to the reference list.
https://www.sciencedirect.com/science/article/pii/S2352340921003991

Yours sincerely,
Thank you.

Experimental design

no comment

Validity of the findings

no comment

Additional comments

no comment

---

## Round 0.3 · accepted · Accept

I am pleased to accept your work. Thank you for submitting to PeerJ, and I look forward to seeing your published version.